# Efficacy of a portable, moderate-resolution, fast-scanning DMA for ambient aerosol size distribution measurements

Stavros Amanatidis[1,2], Yuanlong Huang[1], Buddhi Pushpawela[1], Benjamin C. Schulze[1], Christopher M. Kenseth[1], Ryan X. Ward[1], John H. Seinfeld[1], Susanne V. Hering[2], and Richard C. Flagan[1,*]

[1]California Institute of Technology, Pasadena, CA, USA
[2]Aerosol Dynamics Inc., Berkeley, CA, USA
**Correspondence:** Richard C. Flagan (flagan@caltech.edu)

**Abstract.**

Ambient aerosol size distributions obtained with a compact, scanning mobility analyzer, the "Spider" DMA, are compared to those obtained with a conventional mobility analyzer, with specific attention to the effect of mobility resolution on the measured size distribution parameters. The Spider is a 12-cm diameter radial differential mobility analyzer that spans the 10–500 nm size range with 30 s mobility scans. It achieves its compact size by operating at a nominal mobility resolution $R = 3$ (sheath flow = 0.9 L/min, aerosol flow = 0.3 L/min), in place of the higher sheath-to-aerosol flow commonly used. The question addressed here is whether the lower resolution is sufficient to capture key characteristics of ambient aerosol size distributions. The Spider, operated at $R = 3$ with 30 s up and down scans, was collocated with a TSI 3081 long-column mobility analyzer, operated at $R = 10$ with a 360 s sampling duty cycle. Ambient aerosol data were collected over 26 consecutive days of continuous operation, in Pasadena, CA. Over the 17–500 nm size range, the two instruments exhibit excellent correlation in the total particle number concentrations and geometric mean diameters, with regression slopes of 1.13 and 1.00, respectively. Our results suggest that particle sizing at a lower resolution than typically employed may be sufficient to obtain key properties of ambient size distributions, at least for these two moments of the size distribution. Moreover, it enables better counting statistics, as the wider transfer function for a given aerosol flowrate results in a higher counting rate.

## 1 Introduction

Mobility measurements of atmospheric aerosols in the 10–500 nm size range are important to atmospheric aerosol characterization (McMurry, 2000). Measurements aloft are especially important to understand aerosols in remote regions (Creamean et al., 2020; Herenz et al., 2018), and to mapping three-dimensional profiles (Mamali et al., 2018; Ortega et al., 2019; Zheng et al., 2021). Traditional mobility analyzers that span this size range are large and not suitable for most unmanned aerial vehicle (UAV) or tethered balloon payloads, which increasingly serve as platforms for aerosol characterization aloft. Moreover, aircraft measurements also require a fast scan time resolution to enable a good spatial resolution, as time is proportional to distance traveled in a moving platform.

To that end, Amanatidis et al. (2020) developed the "Spider DMA", a compact, lightweight, and fast differential mobility analyzer (DMA). The instrument was designed for 10–500 nm sizing, with an aerosol flowrate of 0.3 L/min to provide adequate counting statistics on ambient aerosol over the time window appropriate for moving platforms. Its compact size was achieved in part through reduction of mobility resolution. Instead of the typical ratio of sheath-to-aerosol flows of 10, the Spider DMA employs a flow ratio of 3. For given sample flowrate, the most commonly used flowrate ratio of 10 requires a larger sheath flow, which in turn requires a larger mobility analyzer to reach the same maximum particle size.

While high size resolution is important for specific applications, such as in laboratory calibrations that employ a DMA as a calibration aerosol source, it may not be critical for ambient size distribution measurements, wherein the particle distribution spans a much wider size range than the transfer function of the DMA. Lower DMA resolution has also been successfully employed for reconstructing aerosol dynamics process rates in chamber experiments (Ozon et al., 2021). In addition to the smaller physical size of the instrument, operating at lower resolution increases the particle count rate owing to the wider DMA mobility window, thereby reducing measurement uncertainty. This can be an important factor for low-concentration measurements. Moreover, the resulting lower sheath flow requirements enable the usage of more compact and less power-demanding pumps, which further facilitates the overall portability of the instrument.

The question explored in this paper is whether the moderate resolution mobility sizing of the Spider DMA is sufficient to capture the important characteristics of atmospheric aerosol size distributions. We begin with the derivation of the Spider DMA transfer function through a combination of finite element simulations and laboratory calibrations. We then present a field validation by comparison of ambient aerosol data from the new instrument with that obtained from a traditional long-column cylindrical DMA (LDMA) operated at a nominal resolution of $R = 10$ during nearly one month of continuous operation of the two, co-located instruments.

## 2 Methods

### 2.1 Spider DMA

The prototype Spider DMA sizing system consists of the "Spider" DMA (Amanatidis et al., 2020) and the "MAGIC" particle counter (Hering et al., 2014, 2019). The Spider is a compact mobility analyzer designed for applications requiring high porta-bility and time resolution. It features a radial flow geometry and a sample inlet system that distributes the flow azimuthally through curved ("Spider"-like) flow channels. The instrument was designed to operate at 0.3 L/min sample and 0.6–1.2 L/min sheath flowrates, offering size classification in the 10–500 nm size range. Owing to its small classification volume, the mean gas residence time in the classifier is on the order of $\sim 1$ s, making it possible to complete its voltage scan in times well below 60 s without significant smearing of its transfer function.

The "MAGIC" (Moderated Aerosol Growth with Internal water Cycling) particle counter is a laminar-flow water-based CPC. It employs a particle growth tube chamber with three stages (cool, warm, and cool) in which ultrafine particles grow by heterogeneous water vapor condensation to $> 1\,\mu m$, and are subsequently counted by an optical detector. The final stage of the MAGIC CPC growth tube (moderator) recovers excess water vapor, enabling long-term operation without the need of a

reservoir or water refilling. The instrument operates at a sample flowrate of 0.3 L/min, and has a 50% detection cut-point of $\sim 6$ nm.

## 2.2 Transfer function determination by finite element modeling

Amanatidis et al. (2020) evaluated the Spider DMA transfer function in static-mode based on the Stolzenburg (1988) transfer function model and its derivation for radial flow classifiers (Zhang et al., 1995; Zhang and Flagan, 1996). Here, we evaluate its transfer function in "scanning" mobility mode, wherein the electric field is varied continuously in an exponential voltage ramp (Wang and Flagan, 1990). The scanning transfer function was evaluated with 2D finite element COMSOL Multiphysics simulations of flows, quasi-steady-state electric field, and particle trajectories. Simulations were performed for 0.9 L/min sheath and 0.3 L/min aerosol flowrates, scanning voltage in the range 5 – 5000 V, and 30 s exponential ramps for both up- and down-scans. Particles were modeled with the "Mathematical particle tracing" module, in which particle mass was assumed to be negligible since the electric field varies slowly, on a time scale that is long compared to the aerodynamic relaxation time of the particles being measured. Particle trajectories were calculated by assigning particle velocity vector components equal to the steady-state fluid field solution, combined with the axial velocity acquired from interaction with the time-varying electrostatic field. Moreover, a Gaussian random-walk was employed in each time step of the solver to simulate particle Brownian motion, with a standard deviation proportional to particle diffusivity, i.e. $\mathrm{d}\sigma = \sqrt{2\,D\,\mathrm{d}t}$. Monodisperse particles were injected in regular intervals over the scan, varying from 0.025 s for large particles to 0.003 s for those in the diffusing size range to capture in sufficient detail the Brownian motion along the particle trajectories. Modeling was repeated for 10 discrete particle sizes, spanning the dynamic range of the classifier. Details on the Spider DMA geometry employed in the modeling, as well as an example with particle trajectories over the Spider voltage scan are included in the supplementary material (Figures S1 and S2).

## 2.3 Experimental

The two sizing instruments, the Spider DMA and the LDMA system, were operated in parallel, sampling ambient air from a roof top at the Caltech campus in Pasadena, CA. Measurements were made between May 16 – June 11, 2020, and were done as part of a study of the impacts of the COVID-19 pandemic shut-down on air quality. The experimental setup used is shown in Figure 1. Ambient aerosol samples passed through a soft X-ray charge conditioner, and were subsequently split between the two mobility sizing systems, thereby ensuring that the charge status of the aerosols seen by the two instruments was identical. The charge conditioner is a prototype device that was developed recently at Caltech. It is based upon a Hamamatsu soft X-ray source that directly ionizes the air around the incoming aerosol flow. Both DMA systems were operated in scanning mode. Both used a MAGIC water-based CPC as the detector. The size pre-cut stage in the inlet of both CPCs was removed to avoid additional smearing of the transfer functions.

The Spider DMA was operated at 0.9 L/min sheath and 0.3 L/min aerosol flowrates. A piezoelectric blower (Murata, MZB1001T02) was enclosed into a sealed housing to serve as a recirculating pump for the Spider sheath flow. The pump assembly weighs $\sim 60$ g. Operating at a frequency of 24-27 kHz, this pump produces only very small pressure fluctuations that are effectively damped by the capacitance of the downstream filter. With feedback control, the pump attains a steady flow

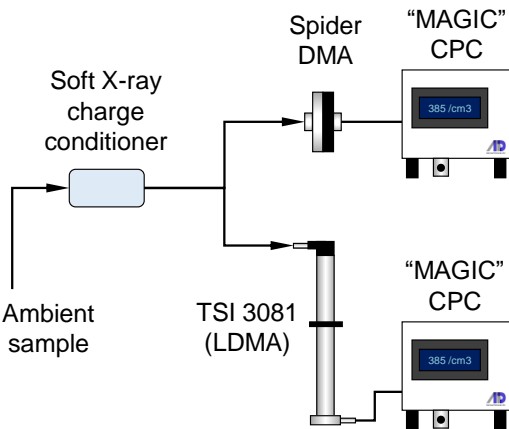

**Figure 1.** Schematic of the experimental setup used to evaluate the Spider DMA. The prototype instrument was operated at 0.9 L/min sheath and 0.3 L/min aerosol flowrates, and a scanning voltage program consisting of a 30 s upscan followed by a 30 s downscan. A TSI 3081 long-column DMA, operated at 3.0 L/min sheath and 0.3 L/min aerosol flows, 240 s upscans, was used for comparison. Both sizing systems used an ADI "MAGIC" CPC as the particle detector.

up to $\sim$ 1 L/min within about 1 s, making it well suited to operating in an environment in which the pressure varies slowly, as in UAV applications. The Spider DMA scanning program included a 30 s upscan followed by a 30 s downscan, during which the electrode voltage was exponentially varied between 5 – 5,000 V. The voltage was held steady for an additional 2 s at each end of the voltage ramp to allow for incoming particles to transmit through the classifier. Particle counts over the scan were recorded with a 5 Hz rate.

The LDMA system was based on a TSI 3081 long-column DMA operated at 3.0 L/min sheath and 0.3 L/min aerosol flowrates, offering classification in the 17–989 nm size range. The scans consisted of an exponentially increasing (upscan) voltage ramp between 25–9,875 V with a 240 s duration. As with the Spider DMA, the LDMA voltage was held constant at the beginning and end of the ramp. Owing to its longer mean flow residence time, the LDMA voltage hold periods were set at 40 s, bringing its overall duty cycle to 360 s. Particle counts for the LDMA system were recorded with a 2 Hz sampling rate. Data acquisition and instrument control (flows, high voltage) was performed with custom LabVIEW software for both systems.

## 2.4 DMA scanning conditions

Comparison of the scanning voltage conditions between the two DMAs requires accounting for differences in geometry, flowrates, and voltage scanning rates. The appropriate non-dimensional quantity that describes the DMA scanning rate is given by $\theta_s = \frac{\tau_{HV}}{t_g}$, the ratio of the exponential voltage ramp time constant, $\tau_{HV}$, to the classifier mean gas residence time, $t_g$. At large $\theta_s$ values, typically $\theta_s > 10$, the rate at which the scanning voltage varies as particles transmit through the classifier is slow, and the transfer function approximates the "static" DMA transfer function. At small $\theta_s$ values, the scanning voltage changes quickly relative to the particle residence time, smearing the transfer function, which becomes pronounced as

$\theta_s$ approaches unity (Russell et al., 1995; Collins et al., 2004). For the Spider DMA operating conditions, $\tau_{HV} = 4.34$ s and $t_g = 1.30$ s, resulting in $\theta_s = 3.35$. For the LDMA, $\tau_{HV} = 40.14$ s and $t_g = 7.52$ s, resulting in $\theta_s = 5.34$. Here, even though $\tau_{HV}$ of the LDMA is about 10 times larger (i.e., slower) than that of the Spider, its dimensionless scanning rate ($\theta_s$) is only about 1.6 times smaller owing to its much longer flow residence time. In absolute terms, the scanning rate employed in both DMAs is moderate.

## 2.5 Data inversion & analysis

Particle size distributions were obtained by inverting the raw particle counts recorded over each voltage scan. Raw counts were smoothed prior the inversion to minimize inversion artifacts. Locally Weighted Scatterplot Smoothing (LOWESS) regression (Cleveland, 1979) was employed for the Spider DMA data with a 10% smoothing window (i.e., 15 data points). The LDMA raw counts were smoothed by applying a moving average filter with a span of 5 data points. The smoothed data were then inverted by regularized non-negative least squares minimization. Tikhonov regularization was used for both systems, with $\lambda = 0.140$ and $\lambda = 0.015$ regularization parameters for the Spider DMA and LDMA data, respectively. Those values were found to provide stable solutions without over-constraining the inversion results.

The inversion kernel for the Spider DMA system was based on the scanning transfer function of the Spider DMA obtained by finite element modeling. In order to generate a dense kernel required for the inversion, the modeled transfer function data were fitted to Gaussian distributions, whose parameters were subsequently fitted to analytical expressions that allowed generation of transfer functions at any instant (i.e., time bin) over the voltage scan. The Spider transfer functions were subsequently convoluted with a continuous stirred-tank reactor (CSTR) model (Russell et al., 1995; Collins et al., 2002; Mai et al., 2018) to take into account the time response of the MAGIC CPC. A 0.35 s time-constant was used for the CSTR model in the Spider DMA system (Hering et al., 2017). The resulting transfer function was combined with a size-dependent transmission efficiency model described by Amanatidis et al. (2020) to take into account particle losses occurring at the Spider inlet, as those are not evaluated in the 2D finite element modeling. Raw counts were shifted to earlier time bins to account for the 1.50 s plumbing time delay between the Spider outlet and the MAGIC CPC detector. Because the simulation enabled a strictly monodisperse "calibration" aerosol, the ratio of the number exiting the DMA during a particular counting time interval to the upstream particle number is the instrument transfer function. The kernel for the LDMA system was based on the scanning transfer function model derived recently by Huang et al. (2020). A CSTR model with a characteristic time of 0.35 s, and a plumbing delay time of 0.95 s were used to incorporate the response of the MAGIC CPC used in the LDMA system.

The Wiedensohler (1988) fit to the Hoppel and Frick (1986) numerical evaluation of the Fuchs (1963) charge distribution has been used in the data inversion. Note that, since both instruments sampled from the same soft X-ray charge conditioner, any deviations from the assumed charge distribution will not affect the comparison between the two instruments.

## 3  Results

### 3.1  Spider scanning transfer function

Figure 2 shows the scanning transfer function of the Spider DMA evaluated by finite element modeling. Results are plotted as
a function of time in the scan, for upscan and downscan voltage ramps. Each peak represents the ratio of particle number at
the outlet to the inlet, for a specific input particle size. Finite element modeling data, shown with symbols, have been fitted to
Gaussian distributions, shown with solid lines, which provide a close approximation to both upscan and downscan modeling
data. As will be shown next, the Gaussian fits are subsequently employed to generate the transfer function at any time instance
over the scan.

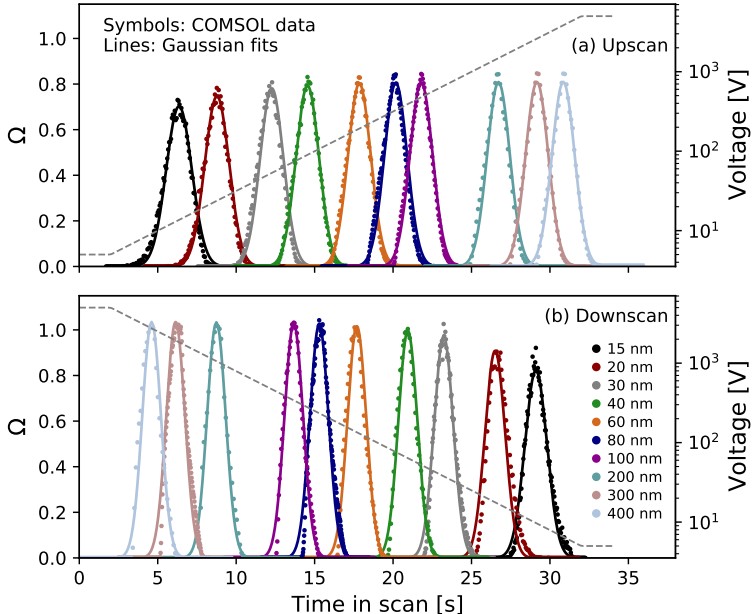

**Figure 2.** Finite element modeling of the Spider DMA scanning transfer function for (a) upscan and (b) downscan exponential voltage ramps
with 30 s duration, 0.9 L/min sheath and 0.3 L/min aerosol flowrates. Symbols correspond to finite element modeling data (ratio of particle
number at the outlet to the inlet); solid lines show Gaussian distributions fitted to the modeling data; dashed lines indicate the scanning
voltage program (values shown on right y-axis).

Comparison between upscan and downscan peaks reveals a distinct difference; downscan peaks have a higher maximum
number ratio. Moreover, they are somewhat narrower than the upscan peaks. It should be noted that the transmission efficiency
through the classification zone of a DMA is proportional to the area under the peak, rather than its maximum value. Hence,
particle transmission over downscans is not necessarily higher than upscans. Here, the area of the Gaussian curves used to fit
the transfer function modeling data was on average ~3.5% larger for downscans than upscans. This difference is likely due
to the slightly asymmetrical shape of the downscan transfer function, which can be observed at the onset (i.e., lower left side)

of each peak in Figure 2b where the fitted curves are somewhat wider than the modeling data. A closer comparison between upscan and downscan fitting parameters is provided in the supplementary material (Figure S3). Diffusional broadening of the transfer function becomes important in the low voltage region of each ramp, increasing the transfer function width as voltage decreases, though the broadening is less than would be seen with a higher resolution DMA (Flagan, 1999).

The differences in the transfer function between upscans and downscans is the result of the scanning voltage operating mode and particle interaction with the boundary flow layers near the DMA electrode walls. Owing to the laminar flow profile, particles near the electrode walls acquire lower velocities than those in middle of the electrode gap. Over downscans, a fraction of the incoming particles interacts with the boundary layer adjacent to the wall that houses the exit slit of the classifier. As voltage drops below a certain threshold, those particles reach the exit of the classifier, albeit with a time delay relative to

particles of the same mobility whose trajectories did not interact with the boundary layer. This results in a particle exit time reallocation, which alters the shape of the downscan transfer function as the voltage drop becomes more rapid. This condition is inhibited over upscans, since the respective boundary layer is formed on the wall opposite to the exit flow, and is exhausted through the excess flow.

        Collins et al. (2004) and Mamakos et al. (2008) demonstrated the impact of scanning voltage on the transfer function of

the cylindrical DMA. Over downscans, the transfer function deviates from the symmetric triangular or Gaussian shape, and becomes skewed. The effect becomes larger for fast scans, and is significant when the effective scan rate $\theta_s < 2$. This is also true for the Spider DMA, as shown in Figure 2b, however the shape distortion is relatively small given the moderate Spider scan rate ($\theta_s = 3.4$). Moreover, in contrast to the cylindrical DMA, the boundary layers in the radial DMA are symmetric, which reduces the downscan distortion compared to its cylindrical counterpart. Over upscans, the width of the scanning transfer

function broadens, but retains its symmetric shape. For this reason, downscan data are often discarded in scanning DMA data analyses, as the more irregular shape of the transfer function is more difficult to parameterize. However, this strategy comes with a penalty in sampling time resolution, owing to the "dead" time associated with the discarded downscan that is required after each upscan. The dead time required depends on the classifier mean gas residence time (typically $> 2$–$3 \times t_g$) and the capacitance of the DMA high-voltage supply. As the Spider DMA scanning transfer function can be described with good

fidelity for both upscans and downscans, both are included in the data analysis to maximize its time resolution.

        Figure 3 shows the integrated transfer function of the Spider DMA system for the same operating conditions as those used in the experiments. The voltage program, shown in Figure 3a, consists of a 2 s hold time at 5 V, followed by a 30 s upscan up to 5000 V, a 2 s hold time at 5000 V, and a 30 s downscan to 5 V. The classified particle size follows roughly the exponential increase and decrease of the voltage over the scan. The peaks shown in Figure 3b consist of the Gaussian approximation

of the Spider transfer function shown in Figure 2, combined with the size and time response of the MAGIC CPC, and the size-dependent transmission efficiency in the Spider inlet (Amanatidis et al., 2020).

### 3.2    Data inversion example

Figure 4 demonstrates an inversion example for representative Spider DMA data. Particle raw counts recorded at each time bin over the upscan and downscan are shown in Figure 4a. Smooth curves are fitted to the raw counts data to minimize artifacts in

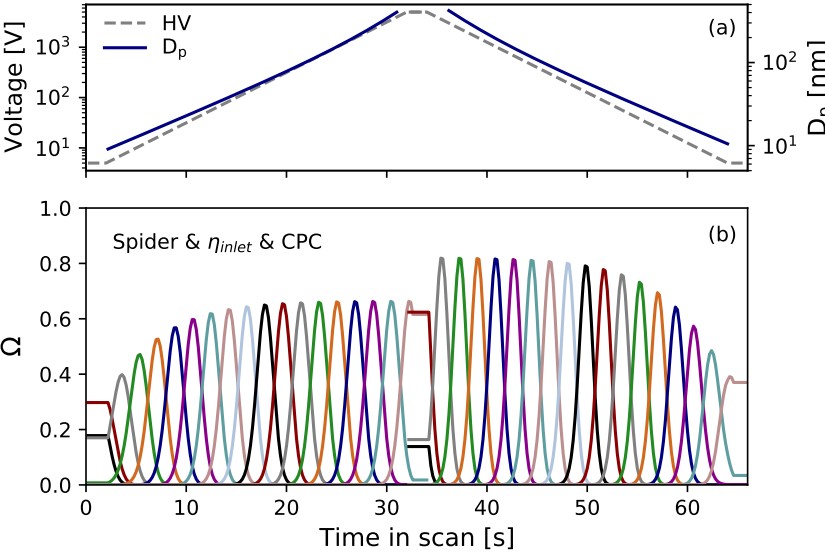

**Figure 3.** a) Scanning voltage and classified particle size over the Spider DMA scan. b) Transfer function of the integrated Spider DMA - MAGIC CPC system (ratio of particle number at the outlet to the inlet), consisting of the Spider DMA scanning transfer function combined with its inlet transmission efficiency and the MAGIC CPC response.

the inversion process. The resulting size distributions, employing an inversion kernel based on the scanning transfer function in Figure 3b, are shown in Figure 4b. Up- and downscan distributions are almost identical in both shape and magnitude. The mean of the two distributions, as shown here, is used as the output of each scan. Overall, considering all measurement data collected in this work, upscan raw counts data inversion yielded distributions with consistent, but slightly higher ($3.7\% \pm 2.3\%$) total particle number than downscans.

## 3.3    Instrument comparison

Figure 5 demonstrates the effect of sizing resolution on the counting rate of the downstream particle detector. As both the Spider DMA and the LDMA operated at the same aerosol flowrate, one would expect a higher counting rate for the Spider DMA system owing to its wider transfer function. Indeed, as shown in Figure 5, this was the case. The data presented here are the average of particle count rates during upscans over an eighteen minute period (corresponding to 3 LDMA upscans, 17 Spider upscans). This example was selected as a representative comparison case since the resulting particle counts distribution is centered near the middle of the overlapping mobility range. The integral of the counting rate with respect to scanning mobility for each instrument (i.e., area below the data points in Figure 5), was larger by a factor of 3.325 in the Spider measurement than the LDMA; this is almost exactly the same as the inverse of the sizing resolution ratio (i.e., 10 / 3) of the two DMAs. In fact, this ratio was rather consistent (within $\pm 10\%$) despite the size distribution variation over the course of the day, corroborating that, for given aerosol flowrate, lower DMA resolution results in higher counting rates, thus enables better counting statistics.

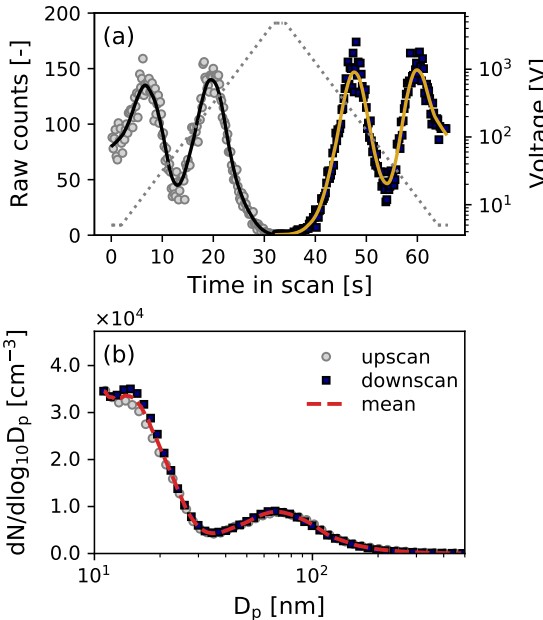

**Figure 4.** Example of Spider DMA data inversion. a) Raw counts per time bin (symbols) recorded over the voltage ramp (up- and down-scan). Solid lines indicate LOWESS smoothing to the raw counts. b) Resulting size distributions after data inversion. The dashed line shows the mean of the up- and downscan distributions.

Figure 6 illustrates an excerpt of the Spider and LDMA size distribution measurements over a time period of 3 days. The two instruments report similar diurnal variation in the particle size distribution, in both size and number concentration. Increased particle concentrations were recorded in the early afternoon of each day, a regular occurrence as particles from morning traffic are transported by the sea breeze from Los Angeles to Pasadena where the measurements took place. Concentrations begin
to drop later in the afternoon and through the evening, from about 15,000 cm$^{-3}$ to below 5,000 cm$^{-3}$. The geometric mean diameter (GMD) of the size distribution ranged between about 30–60 nm, and was smaller over the high number concentration events recorded in early afternoon.

Figure 7 shows the evolution of the size distribution over a period of 2 hours in the afternoon of May 28, 2020 (indicated with dashed box in Figure 6d), measured with the Spider and the LDMA system. Since the measurement duty cycle of the two
instruments was different (66 s for the Spider vs 360 s for the LDMA), we employed 30 min averaging of the recorded size distributions. This corresponds to 5 scans for the LDMA, and about 27 up- and down-scans for the Spider. The shaded areas of the averaged distributions represent the variation over the averaging period. Starting from a mono-modal distribution with a peak at $\sim 45$ nm (panel a), the size distribution transitioned to a bi-modal one over a period of 60 min (panels b, c), before transitioning back to a mono-modal distribution (panel d). As indicated by the shaded areas, there was high variation in the
aerosol concentration during this transition event. Overall, the measurement of the two instruments was in good agreement both in terms of sizing and concentration, suggesting that the lower sizing resolution in the Spider DMA was adequate in capturing

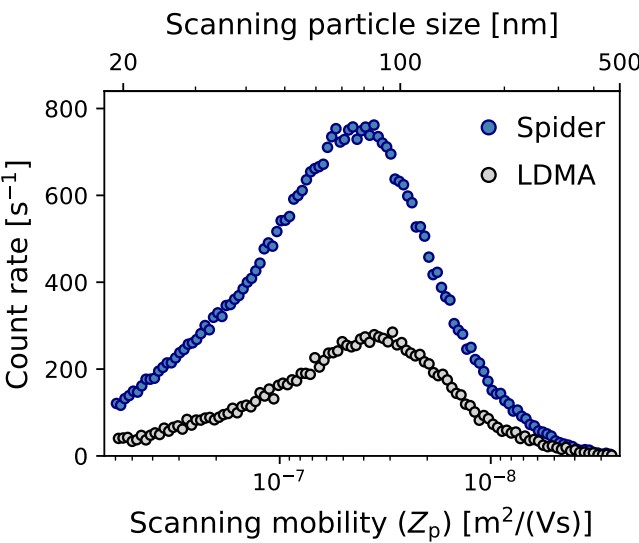

**Figure 5.** Sizing resolution effect on the particle count rate of the Spider DMA (R=3) and LDMA (R=10) systems. Data shown are the average of raw particle count rates during upscans over an eighteen minute period (corresponding to 3 LDMA upscans, 17 Spider upscans) measured in the morning of June 1, 2020. Both systems operated at 0.3 L/min aerosol flowrate.

the details of the size distribution. An animation video with side-by-side comparison of 30-min averaged distributions for the entire testing period is included in the Supplementary Material (Amanatidis et al., 2021).

Figure 8 compares the total number and geometric mean diameter measured by the two instruments over the entire testing period. Each data point corresponds to a 1-hour average of the size distribution measured by each instrument, calculated over the 17–500 nm size range where the two systems overlap. Overall, the comparison includes 550 h of measurement data. In order to identify outliers in the data, we employed the "RANSCAC" (random sample consensus) algorithm (Fischler and Bolles, 1981). In this, random samples of the data are selected, analyzed, and classified as inliers and outliers through an iterative routine. The outliers identified are shown in Figure 8 with open square symbols.

Next, a linear regression model (no intercept) was fitted to the data (excluding outliers) to evaluate the correlation between the two instruments. Since both instruments include measurement errors, we employed Orthogonal Distance Regression (Boggs et al., 1987), where errors on both the dependent and independent variable are taken into account in the least squares minimization. The resulting regression lines exhibit slopes of $\alpha = 1.13$ and $\alpha = 1.00$ for number concentration and GMD, respectively, suggesting an overall excellent agreement between the instruments. Moreover, Pearson correlation coefficients of $\rho = 0.98$ and $\rho = 0.93$ indicate a strong correlation for both metrics of the size distribution.

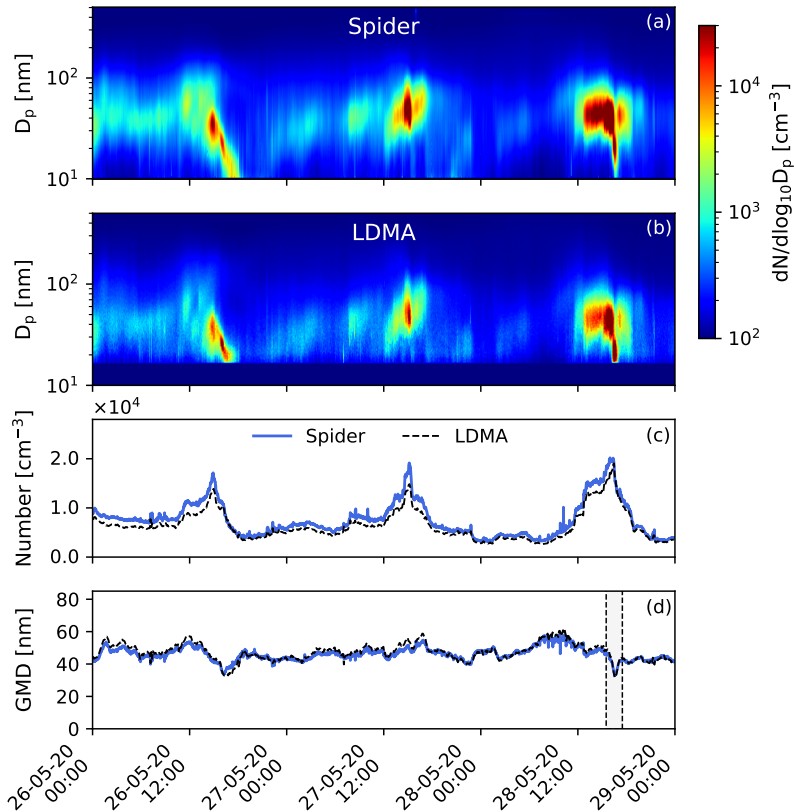

**Figure 6.** Evolution of the particle size distribution over a period of 3 days measured by a) the Spider DMA, and b) the LDMA system. Corresponding total particle number and geometric mean diameter, calculated over the 17–500 nm size range, are shown in panels (c) and (d), respectively. Solid blue color in panel (b) (size range <17 nm) was used for no available data in the LDMA system. The dashed box in panel (d) indicates the time period shown in Figure 7.

### 3.4 Operational observations

The prototype Spider DMA used in this study incorporated an electrostatic-dissipative plastic that failed after several months of continuous operation, causing arcing within the instrument at the highest voltages. The Spider DMA has been redesigned to eliminate this material, and is currently being tested. This new Spider DMA has relatively minor changes to the classification region of the prototype presented here, and employs the same moderate resolution approach to maintain a compact size.

### 4 Summary & conclusions

We evaluated the performance of the Spider DMA, a highly-portable particle sizer, in measuring ambient size distributions against a co-located particle sizer based on a TSI 3081 long-column DMA (LDMA). Comparison measurements were per-

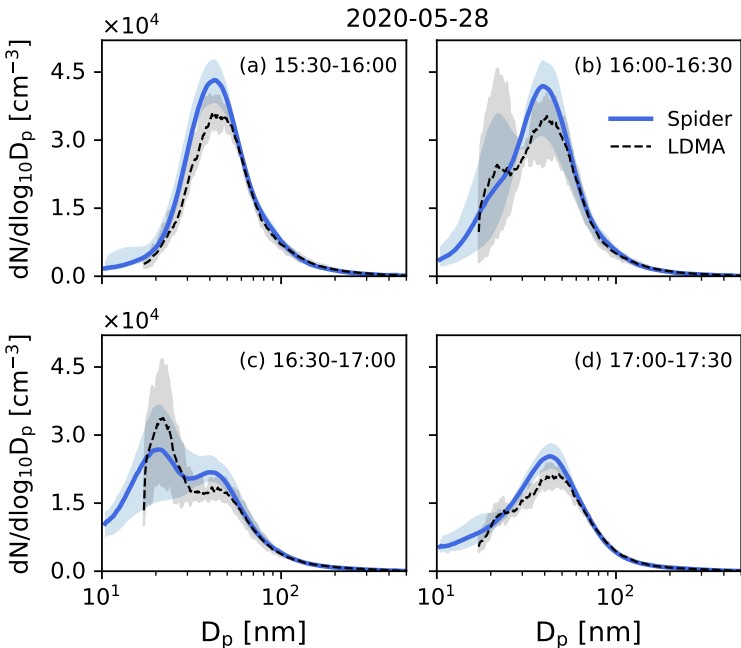

**Figure 7.** Evolution of the size distribution in the afternoon of May 28, 2020, as measured by the Spider and LDMA systems. Lines represent the mean of size distributions measured over a period of 30 min. Shaded areas demonstrate the variation of the size distribution over the averaging period, indicating maximum and minimum values.

formed at the Caltech campus in Pasadena, CA over a period of 26 days, between May 16 – June 11, 2020, as part of a field
campaign examining the effects of COVID-19 shut-down on air quality. The Spider DMA system was operated at a lower
nominal sizing resolution (0.9 L/min sheath and 0.3 L/min aerosol flowrates, $R = 3$) than the LDMA (3.0 L/min sheath and
0.3 L/min aerosol flowrates, $R = 10$), and at a higher time resolution (30 s vs 240 s scans).

The transfer function of the Spider DMA was obtained by finite element modeling at the conditions employed in the experiment, which included both up- and downscan exponential voltage ramps with 30 s duration. Owing to the Spider radial flow
geometry and short gas flow residence time, distortion of the downscan transfer function shape is minimal at the scan rates
employed, enabling usage of both upscan and downscan data, thereby increasing time resolution. Modeling data were fitted
to Gaussian distributions, and were combined with the experimentally-determined transmission efficiency of the Spider DMA
and the MAGIC particle counter response function to generate the inversion kernel of the combined system. Data inversion of
the LDMA system was based on the semi-analytical model of the LDMA scanning transfer function derived by Huang et al.
(2020).

Regression analysis of 550 h of measurement data showed an overall excellent correlation between the two instruments, with
slopes of $\alpha = 1.13$ and $\alpha = 1.00$, and Pearson correlation coefficients of $\rho = 0.98$ and $\rho = 0.93$ in the reported particle number
and geometric mean diameter (GMD), respectively. The present results suggest that two key characteristics of ambient size

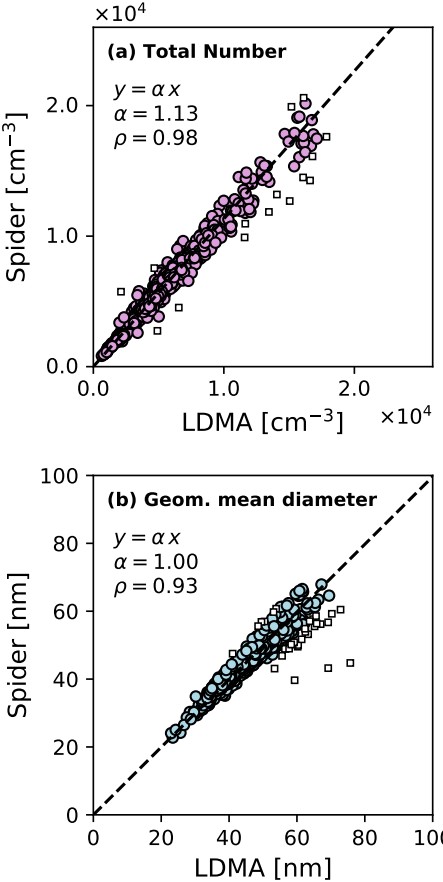

**Figure 8.** Comparison of a) total particle number, and b) geometric mean diameter, measured by the Spider and LDMA systems over a period of 26 days of continuous testing. Each point represents 1 hour averaged data, calculated over the 17–500 nm size range where the two instruments overlap. Square symbols show outliers excluded from the regression analysis. Dashed lines represent a linear regression model (no intercept) fitted to the data. $\rho$ values denote the Pearson correlation coefficient between the measurement data of the two instruments.

distributions, geometric mean diameter and number concentration, are sufficiently captured when operating the DMA at lower
resolution than is typically employed. Moreover, use of lower resolution, where appropriate, has several distinct advantages.
For the same aerosol flow rate and range in particle mobilities, reducing the nominal resolution reduces the required sheath flow
and hence reduces the physical size of the DMA. In turn, this reduction in physical size at the same aerosol flow rate reduces the
residence time within the classification region, enabling faster scans. Additionally, for the same aerosol flow, the wider mobil-
ity window increases the particle count rate, thereby improving measurement statistics. While some applications may require
higher resolution, this study demonstrates the efficacy of lower resolution measurements for ambient aerosol characterization,
and illustrates the commensurate advantages of faster measurements in a smaller package.

*Author contributions.* SA performed the finite element modeling for the Spider DMA instrument, analyzed its measurement data, generated the figures, and wrote the manuscript text. YH analyzed the LDMA instrument data and prepared the experimental setup. BP, BCS, CMK and RXW collected the measurement data and provided technical maintenance to the instruments. JHS reviewed and provided editorial feedback on the manuscript. SVH and RCF planned the experiments, and contributed to results interpretation and editing of the manuscript.

*Competing interests.* RCF and SA are inventors of the "Spider" DMA technology patent (US10775290B2) which is licensed to SVH's company. The rest of the authors declare that they have no conflict of interest.

*Disclaimer.* Neither the United States Government nor any agency thereof, nor any of their employees, makes any warranty, express or implied, or assumes any legal liability or responsibility for the accuracy, completeness, or usefulness of any information, apparatus, product, or process disclosed, or represents that its use would not infringe privately owned rights. Reference herein to any specific commercial product, process, or service by trade name, trademark, manufacturer, or otherwise does not necessarily constitute or imply its endorsement, recommendation, or favoring by the United States Government or any agency thereof. The views and opinions of authors expressed herein do not necessarily state or reflect those of the United States Government or any agency thereof.

*Acknowledgements.* The authors gratefully acknowledge support by the U.S. Department of Energy, Office of Science, under Award Number(s) DE-SC0013152, by the Department of Health and Human Services, Centers for Disease Control and Prevention under Award OH010515

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
