# Peer review of "Efficacy of a portable, moderate-resolution, fast-scanning DMA for ambient aerosol size distribution measurements"

_Atmospheric Measurement Techniques, 2021_

## Author Response (AR1)

**Reviewer #1**

The authors would like to thank the reviewer for their constructive feedback on this paper. Our point-to-point response to the reviewer's comments is listed below.

The manuscript by Stavros Amanatidis et al. entitled "Efficacy of a portable, moderate-resolution, fast-scanning DMA for ambient aerosol size distribution measurements" reports an intercomparison of a novel SEMS or MPSS system consisting of the "SPIDER" DMA and a "MAGIC" CPC. The main question raised in the paper is whether this system operated a relatively low DMA resolution is able to catch the key characteristics of ambient aerosol size distributions.

The paper comprehensively evaluates the transfer function of the "Spider DMA", an also reports field measurement data showing very good agreement between the novel system and a more traditional scanning DMA system.

From this reviewer's point of view, the paper is very welcome to be published in "Aerosol Measurement Techniques" although it should not be accepted for publications until following remarks have been considers in a revised version of the manuscript:

1)    P1/L18ff

The authors state that traditional mobility analyzers are large and most often not suitable for UAV, but that the "Spider DMA" would be appropriate to be used on moving platforms.

It would be helpful if the authors could also elaborate on how the sheath air flow supply of the Spider DMA would look like when used on n UAV or other moving platform – especially when compared to "traditional" sheath flow supplies.

Indeed, the low flow requirements of the Spider DMA enable the usage of more compact and low-power pumps. While there are more than one options that would be appropriate for the Spider sheath flow, we included in the "Methods/Experimental" sub-section a brief description of the pump used in this prototype system, which is an in-house prototype pump based on a low-power piezoelectric micro-blower. The pump assembly weight is ~ 60 g.

2)    P3/L74 ff.

The authors report the use of a soft X-ray charge conditioner. It would beneficial to specify brand / make of the specific instrument. It is not the objective of this paper, but nowadays soft x- ray ionizers are often used for size distribution measurement, without knowing the actual charging probabilities or knowing if the Fuchs charging theory is applicable without any adaptions. Therefore, at least the type of the used instrument should be stated.

We used a prototype soft X-ray charge conditioner that was developed recently at Caltech. It is based upon a Hamamatsu soft X-ray source. Detailed calibration of the charger was stopped by the COVID-19 shutdown, and further delayed when the Spider DMA was deployed to make the measurements reported in this paper. Details of the charger design and a full calibration will be reported in a separate paper. As noted by Steiner and Reischl

(2012), and Leppä et al. (2017), the charge distribution depends upon trace gases in the aerosol sample and may differ from the results of those earlier simulations. We agree with the reviewer that this is critical information for electrical mobility measurements of particle size distributions, but, since both instruments sampled the aerosol from the same charge conditioner, the conclusions drawn from the comparison presented in this paper are not affected. This note was included in Section 2.5 of the revised manuscript.

3) P5/L107:

It should be named "Fuchs charge distribution"

We revised this paragraph to the following:

*"The Wiedensohler (1988) fit to the Hoppel and Frick (1986) numerical evaluation of the Fuchs (1963) charge distribution has been used in the data inversion. Note that, since both instruments sampled from the same soft X-ray charge conditioner, any deviations from the assumed charge distribution will not affect the comparison between the two instruments."*

4) P5/L114 ff

The fact that down-scan peaks have a higher maximum number ratio and are also narrower than the up-scan peaks confuses me. Typically, one would expect the opposite, where the down-scan transfer function also often exhibits a distortion or tail. Therefore, the – as far as this reviewer can say – the most common way would be to use the up-scan data for scanning DMA data rather than the down-scan data.

It would be extremely important for a clear understanding – especially for non-DMA expert readers - to elaborate in more detail on this topic and explain the differences between the down-scan/up-scan transfer functions of the Spider DMA vs. traditional DMAs.

Downscan data are often discarded from scanning DMA analyses because the shape of the transfer function is skewed, and hence more difficult to parameterize. This is true for both cylindrical and radial DMAs. However, the extent of smearing depends on how "fast" or "slow" the scan is relative to the gas mean residence time in the classifier. Collins et al. (2004) and Mamakos et al. (2008) demonstrated the shape of upscan vs downscan transfer functions, for a range of conditions for the cylindrical DMA geometry. Moderate downscan rates result in relatively small distortion in the transfer function, which is also true for the Spider DMA. In those cases, the downscan data are certainly usable. Including the downscan data improves the time resolution in scanning DMAs, which is important for some applications such as moving platform deployments.

We discuss the above in more detail in the revised paper (Section 3.1). Moreover, we included a new subsection (2.5) under "Methods" that provides some background on the scanning conditions of the two instruments used in this work.

**Reviewer #2**

The authors would like to thank the reviewer for their constructive feedback on this paper. Our point-to-point response to the reviewer's comments is listed below.

The authors present the application of the "Spider"-DMA to ambient measurements, demonstrating that despite the lower resolution compared to typical SMPS/DMPS systems, the characteristics of the size-distribution can be retrieved with high accuracy making the "Spider"-DMA a suitable instrument for lightweight particle size-distribution measurements. The article is written in a concise style, the presented Figures are of high quality and the conclusions are scientifically sound. The topic is certainly of interest for the community, as the interest in e.g. unmanned aerial vehicle measurements is increasing due to the usage of drones, where this device could come in handy. I can recommend publication in Atmos. Meas. Techn. subject to some minor revisions:

- 2, l.26-28: Doesn't this also come with the benefit of requiring smaller sheath-flow supplies?

  This is correct; we have included a short discussion in the revised Introduction.

- 3, l.80: Standard SMPS systems often use particle counters with 1 lpm flow rate. This would increase the counting statistics compared to the spider DMA. Does the reduced resolution still outweigh the benefit of an increased sample flow in terms of counting statistics?

  Considering a DMA with given geometric characteristics, the counting rate of the downstream CPC should be proportional to the ratio of aerosol flowrate-to-sizing resolution ($Q_a$ / R). Thus, 10 / 1 lpm sheath / aerosol flow conditions (R=10) would be equivalent to 0.9 / 0.3 lpm conditions (R=3) in terms of counting statistics (both have $Q_a$ / R = 0.1).

- 3, l.84: Why is a different sampling rate used for the LDMA system and the Spider system?

  Different sampling rates were employed due to different scan durations in the two DMA systems. In both cases the sampling rate was sufficient to capture the raw counts variation during the scan.

- 4, l.90: Please be more specific on the inversion algorithm. Tikhonov regularization? What was the choice for regularization parameter? Was the same method applied to the LDMA?

  We have used Tikhonov regularization for both instruments. Additional details on the data inversion are included in the revised manuscript (Section 2.5).

- 5, l.117: Please be more specific where this difference comes from. What's the difference between downscan and upscan voltage operating mode which explains these discrepancies? Figure 4 also shows that there are slight changes in the

inverted distribution. Is this happening repeatedly? And if one scan gives more precise results, why using an up- and downscan procedure instead of one direction only?

We included a discussion in the revised manuscript (Section 3.1) on the differences involved between upscans and downscans. Regarding the inverted data, analysis of all the data collected over the testing campaign showed that upscan and downscan inverted distributions were overall consistent, with upscans yielding slightly higher (~3.5%) total particle number than downscans. The advantage in using both up- and downscan data is the resulting improvement in time resolution by eliminating the time required to return to the starting voltage of a single-direction scan. The accuracy of the inverted data depends on the accuracy of the transfer function model being employed. For moderately slow downscans, this can be realized with good accuracy.

- 6, l.129: It would be very helpful to see the same for the LDMA in order to demonstrate the higher counting statistics provided by the spider DMA. What is the advantage we gain in "counts" compared to the other device?

We added a new figure in the revised paper (Fig. 5) to demonstrate the impact of resolution on DMA counting rate.

Figure 5: Compared to the LDMA there seem to be some spikes in the reconstructed total number in the Spider DMA which do not appear in the LDMA? On the contrary, the contour plot clearly shows more scatter for the inverted LDMA size-distribution. Is this caused by the lower counting statistics (see previous comment) or by a different inversion algorithm (requiring less smoothing)?

Those differences mainly arise due to the different time resolution of the two instruments. For each LDMA scan shown in the contour plot (every 6 min), there are about 6 Spider scans reported. The "spikes" that appear in the Spider data reflect the aerosol variation at those faster scans, which were not captured completely by the lower LDMA time resolution. In fact, a closer look at the number traces shows that, for the majority of these events, there is also a corresponding "spike" in the LDMA number trace, albeit typically weaker. A 6-min moving average in the Spider data would have resulted in a more "rounded" particle number time series.

- Figure 6: Similar to the above comment, the spider size-distribution looks extremely "smooth" here. What is causing this?

The Spider distribution appears smoother because of the combined effect of two factors: a) more smoothing was added in the Spider inversion compared to the LDMA (details included in the revised manuscript – Section 2.5); b) the 30-min average distributions shown in Fig. 6 include about 6 times more Spider scans than the LDMA (i.e., 27 vs. 5 scans), which results in a "smoother" mean distribution over the same time interval.

- 11, l.185: Are the GMD and total number the only key characteristics of a size-distribution? I would be a bit more defensive with that statement.

We have revised our statement to the following:

*"The present results suggest that two key characteristics of ambient size distributions, geometric mean diameter and number concentration, are sufficiently captured when operating the DMA at lower resolution than is typically employed."*

- 11, l.185: Related to the fact that lower resolution is perfectly suitable in reconstructing aerosol formation and growth rates there is a recent paper in Atmos. Chem. Phys. Discuss. by Ozon et al. (https://doi.org/10.5194/acp-2021-99), showing that a wider resolution can indeed help to reconstruct aerosol dynamics process rates. Even if it is at preprint stage, it could be mentioned here as it is making a similar case that size-distribution reconstruction in ambient or chamber experiments does not necessarily require high resolution.

We thank the reviewer for the suggestion; we have added the reference in the Introduction of the revised paper.

**Reviewer #3**

The authors would like to thank the reviewer for their constructive feedback on this paper. Our point-to-point response to the reviewer's comments is listed below.

The authors provide a concise and well reasoned analysis of a relatively low resolution spider DMA system as a means to balance classifier size and robustness of data capture. The analysis of the transfer functions were performed by steady state laminar CFD analysis, which is appropriate for this system (done in separate study). The modeled results are in line with findings from other studies and give a quantified potential for weight and resolution tradeoffs. As a whole the article is worth of publication and should be accepted once addressing the following comments.

My largest disappointment in the article is that it did not engage more broadly with the larger question of what are the theoretical and practical tradeoffs that allow classifiers (in this case DMAs) to be optimized for specific applications. This work gives one data point as to a classifier that works well for the desired application, but does not give the reader insights into whether the system could be further optimized by tradeoffs. A particularly useful inclusion would be general scaling laws that might guide others in the field who are seeking to develop custom classifiers for small (either volume or mass limited) applications.

We agree with the reviewer that this is an important question that needs to be addressed in more detail. This present paper, however, focuses on reporting the efficacy of the moderate-resolution Spider DMA in measuring ambient size distributions, rather than the process of DMA design & optimization, which was (in part) presented in a previous

publication (Amanatidis et al, 2020). Thus, even though this is indeed a topic of interest for the authors, it is outside of the scope of this present work.

Minor points for the article are as follows:

Line 55: Despite the work being based on a previous publication (Amanatidis, 2020) it would be useful to have a schematic in Sec 2.2 to depict basic model parameters and methodology, such as labeled boundary conditions and a few rudimentary results.

We have included additional details on the finite element modeling in the supplementary material, including a schematic of the Spider geometry used in the modeling (Figure S1), as well as a figure with particle trajectories over the voltage scan (Figure S2).

Line 68: It was unclear why the intervals for injection were chosen and what impact it had on the simulation results.

Shorter injection interval (i.e., larger number of particles simulated over the scan) was employed for smaller particles to capture in sufficient detail the additional (Brownian) motion along the trajectories of those diffusive particles.

Line 107: Why was Wiedensohler approximation chosen for an x-ray charger when the ion properties of soft x-ray have been shown to be different than Wiedensohler's results which are calibrated to radioactive neutralizers?

We agree with the reviewer that the Wiedensohler approximation employed might differ from the actual charge probability generated by the soft X-ray charge conditioner. The device used here was a prototype charge conditioner that was developed recently at Caltech, but has not yet been fully characterized. We employed the Wiedensohler correlation as an approximation to the actual charge distribution. Since both instruments sampled the aerosol from the same charge conditioner, the conclusions drawn from the comparison presented in this paper are not affected.

Line 116: It would be nice to have the maximum number ration and "narrower" transfer functions discussed in quantified terms.

We present a comparison between the parameters of the upscan and downscan transfer functions in the supplementary material, Figure S3.

Fig. 3b: I like the visual representation of the scans in Fig 3b, but a quantitative figure would also be useful depicting higth and width of the transfer function.

Figure S3 in the supplementary material presents a more quantitative comparison between the height, width, and area of the upscan and downscan transfer functions.

Line 130: how are the smooth lines fit to the data? What are the smoothing parameters?

We included additional details on the smoothing employed to the raw counts data in the revised manuscript (Section 2.5).

Fig. 6: The smaller modes of the distributions appear to be less distinct in the right plots making them indistinguishable. I would appreciate the authors view of these findings.

Due to the variability of the size distribution over this transient event, and the different time resolution of the two instruments, it is not straightforward to make a conclusive comparison that would explain such subtle differences. We have, however, investigated whether the low resolution of the Spider DMA could potentially be a limitation in capturing bimodal distributions similar to those shown in Figure 6; based on our preliminary analysis, this was not the case. This is also supported by observing the shaded light blue area in panel b, that represents the variability of the size distribution over the averaging interval, which shows that the Spider data included scans where the 1st mode of the distribution was distinguishable.

Typesetting – the authors provide correct spacing between the number and engineering unit, e.g. L/min, in most cases, but fail to provide spaces between numbers and seconds or volts in several places.

Corrected in the revised manuscript.